# Microglia show differential transcriptomic response to Aβ peptide aggregates ex vivo and in vivo

Karen N McFarland[1,2,3], Carolina Ceballos[2,3,4], Awilda Rosario[2,3,4], Thomas Ladd[2,3,4], Brenda Moore[2,3,4], Griffin Golde[2], Xue Wang[5], Mariet Allen[6], Nilüfer Ertekin-Taner[6,7], Cory C Funk[8], Max Robinson[8], Priyanka Baloni[8], Noa Rappaport[8], Paramita Chakrabarty[2,3,4], Todd E Golde[2,3,4]

Aggregation and accumulation of amyloid-β (Aβ) is a defining feature of Alzheimer's disease pathology. To study microglial responses to Aβ, we applied exogenous Aβ peptide, in either oligomeric or fibrillar conformation, to primary mouse microglial cultures and evaluated system-level transcriptional changes and then compared these with transcriptomic changes in the brains of CRND8 APP mice. We find that primary microglial cultures have rapid and massive transcriptional change in response to Aβ. Transcriptomic responses to oligomeric or fibrillar Aβ in primary microglia, although partially overlapping, are distinct and are not recapitulated in vivo where Aβ progressively accumulates. Furthermore, although classic immune mediators show massive transcriptional changes in the primary microglial cultures, these changes are not observed in the mouse model. Together, these data extend previous studies which demonstrate that microglia responses ex vivo are poor proxies for in vivo responses. Finally, these data demonstrate the potential utility of using microglia as biosensors of different aggregate conformation, as the transcriptional responses to oligomeric and fibrillar Aβ can be distinguished.

## Introduction

Alzheimer's disease (AD) is characterized by two hallmark pathologies, senile plaques containing amyloid-β (Aβ) aggregates and neurofibrillary tangles (NFTs) composed of hyperphosphorylated and aggregated τ. Amyloid plaques are the earliest manifestations of the disease process and can appear up to 20 yr before the onset of cognitive symptoms (Bateman et al, 2012). Amyloid pathology, in the absence of τ or neurodegenerative pathology, defines preclinical AD and is the first step along the Alzheimer's continuum in humans (Vickers et al, 2016; Jack et al, 2018; Cummings, 2019). In longitudinal studies, amyloid deposition precedes τ accumulation which is more closely tied to cognitive decline relative to amyloid (Villemagne et al, 2013; Hanseeuw et al, 2019). Furthermore, genetic data strongly support a causal, triggering role for aggregation and accumulation of Aβ in AD (Kunkle et al, 2019)—including the well-studied APOE4 risk allele in late-onset AD, which reduces the clearance of Aβ from the brain (Liu et al, 2013). Yet, despite intensive study, the precise mechanism by which accumulation of Aβ aggregates trigger the degenerative phase of the disease is not well understood.

As the primary immune and phagocytic cell in the brain, the role of microglia has been of growing interest in AD and other neurodegenerative disorders. "Resting" microglia, which constitute up to 10% of the brain, constantly sample the surrounding brain microenvironment and can rapidly respond to an insult (Aguzzi et al, 2013). In AD, the presence of increased "reactive" microglial cells both around senile plaques and in areas of neurodegeneration is a well-established pathological feature (Dickson et al, 1988; Perlmutter et al, 1992; Dickson, 1997). Notably, $A\beta_{42}$ fibrils and oligomers cause microglia activation resulting in the release of proinflammatory cytokines which may contribute to neurotoxicity (Jimenez et al, 2008; He et al, 2012; Dewapriya et al, 2013; Wang et al, 2016). Alterations in microglial activation states can also impact both amyloid and τ pathology in varying ways that are dependent on both the stimulus, the model system, and the pathology that is being assessed.

Over the last decade, a series of genetic studies has firmly linked microglial function to AD. Genetic studies of familial and late-onset AD implicate a large number of loci that contain immune genes in mediating the risk for AD (Harold et al, 2009; Lambert et al, 2009, 2013; Guerreiro et al, 2013; Jonsson et al, 2013; Jin et al, 2015; Carrasquillo et al, 2017; Sims et al, 2017; Kunkle et al, 2019). Furthermore, genetic studies identifying coding variants in three microglial-specific genes (PLCG2, ABI3, and TREM2) highlight the important role that microglia play during neurodegeneration (Guerreiro et al, 2013; Jonsson et al, 2013; Jin et al, 2014; Bellenguez

---

[1]Department of Neurology, University of Florida, Gainesville, FL, USA   [2]Center for Translational Research in Neurodegenerative Disease, University of Florida, Gainesville, FL, USA   [3]McKnight Brain Institute, University of Florida, Gainesville, FL, USA   [4]Department of Neuroscience, University of Florida, Gainesville, FL, USA   [5]Department of Health Sciences Research, Mayo Clinic, Jacksonville, FL, USA   [6]Department of Neuroscience, Mayo Clinic, Jacksonville, FL, USA   [7]Department of Neurology, Mayo Clinic, Jacksonville, FL, USA   [8]Institute for Systems Biology, Seattle, WA, USA

Correspondences: knmcfarland@ufl.edu; tgolde@ufl.edu

et al, 2017; Sims et al, 2017; Conway et al, 2018; van der Lee et al, 2019; Strickland et al, 2020). In addition, system-level data analysis of spatial, single-cell, single-nuclei, and bulk RNA-sequencing (RNA-seq) studies reveal perturbations in immune transcriptional networks as well as distinct subpopulations of microglia that are perturbed in the AD brain (Keren-Shaul et al, 2017; Krasemann et al, 2017; Conway et al, 2018; Friedman et al, 2018; Hammond et al, 2019; Li et al, 2019; Chen et al, 2020; Olah et al, 2020).

The study of microglial cells is challenging in that they are highly responsive to external stimuli and rapidly alter their phenotype once removed from the brain (Bennett et al, 2016). Indeed, system-level transcriptomic studies show that primary microglial cells are poor proxies for in vivo microglia (Butovsky et al, 2014). Even rapid isolation of microglial and subsequent "omic" analyses can be challenging as it is clear the isolation process is sufficient to induce some transcriptional—and likely functional changes. Nevertheless, many labs—including our own—study primary microglial cells in culture. In particular, the application of exogenous Aβ aggregates to microglia is a widely used methodology to study both how microglial respond to Aβ and how effectively the microglia can phagocytose and degrade Aβ.

Here, we used RNA-seq to examine the systems level response of primary microglia in culture to synthetic $A\beta_{42}$ aggregates in either oligomeric (oAβ) or fibrillar (fAβ) form. Our analyses of the transcriptomic data show that microglial cells in culture show massive transcriptional changes when challenged with $A\beta_{42}$ aggregates. Although some of the differentially expressed genes (DEGs) in response to the different forms of $A\beta_{42}$ are altered similarly, many show differential expression in response to oAβ or fAβ. We also compared this global transcriptional response with $A\beta_{42}$ in primary microglial cells in culture to transcriptomic data from a mouse model of amyloid deposition—the amyloid precursor protein (APP) transgenic CRND8 mouse—at 3–20 mo of age (Chishti et al, 2001). Subsequent comparisons of these datasets indicate that most Aβ transcriptional responses in microglia are largely not replicated in the intact brain. This comparison demonstrates that the transcriptional response to Aβ in primary cultures poorly reflect the response to Aβ by microglial cells in the mouse brain. These data amplify the message of several other recent studies indicating that one must be very cautious when using primary microglial cells cultured in isolation to infer mechanistic insights about microglial function in vivo.

# Results

### Large transcriptomic changes in primary microglia after Aβ treatment

Preformed oligomeric (oAβ) or fibrillar (fAβ) forms of $A\beta_{42}$ peptide were applied to primary microglia cultures for 1- or 12-h. oAβ and fAβ were characterized by Western blot and a representative image is shown (Fig S1) demonstrating differences in the high molecular weight species between oligomeric and fibrillar Aβ preparations. After treatment, RNA was isolated and sequenced to identify transcriptional changes in primary microglia that are responsive to different conformations of $A\beta_{42}$ peptide. As noted in the methods, these data along with the mouse CRND8 RNAseq data are publicly available and can be viewed using an interactive data portal. Using cutoff values of a $P$-value (adjusted for multiple comparisons) ≤0.05 and an absolute $log_2$ fold change of 0.5, we

identified acute transcriptional changes after fAβ application after just 1-h (versus control) with 997 up-regulated and 960 down-regulated genes (Fig 1A and Table S1). Gene ontology (GO-MF) and Kyoto Encyclopedia of Genes and Genomes (KEGG) pathway analysis identified down-regulated genes as being enriched in cytoskeletal and extracellular matrix organization (i.e., tubulin binding, motor activity, and extracellular matrix structural constituent; Fig 1E and Tables S2 and S4), whereas up-regulated genes were involved in immune system responses (i.e., receptor for advanced glycation endproducts [RAGE] receptor binding and chemokine activity), and kinase activity (i.e., mitogen activated protein [MAP] kinase phosphatase activity; Fig 1F and Tables S3 and S5).

After 12-h of fAβ treatment, we identified 1,755 up-regulated and 1,975 down-regulated genes when compared with control (Fig 1B and Table S6). Gene ontology (GO) and KEGG pathway analysis revealed enrichment of down-regulated genes involved in cytoskeletal and extracellular matrix organization (i.e., tubulin binding, motor activity, and extracellular matrix structural constituent) in addition to heparin binding and glycosaminoglycan binding (Fig 1E and Tables S8 and S10). Genes up-regulated after 12 h fAβ treatment were enriched in genes involved with antigen processing (transporter associated with antigen [TAP] binding) and proteolytic activity (i.e., endopeptidase activator activity and threonine-type peptidase activity; Fig 1F and Tables S7 and S9).

We next examined the effect of a 12-h oAβ treatment (versus control) on primary microglial cultures. We identified 1,608 up-regulated and 1,394 down-regulated genes after 12 h of oAβ (Fig 1C and Table S11). GO and KEGG pathway analysis revealed that down-regulated genes are primarily involved in DNA transcription (i.e., DNA-binding transcriptional repressor activity and transcription cofactor binding; Fig 1C and E and Tables S13 and S15). Genes up-regulated by oAβ treatment are enriched with GO terms suggestive of cell cycle involvement (i.e., anaphase-promoting complex binding and kinetochore binding; Fig 1F and Tables S12 and S14). A number of the top GO category hits overlap somewhat between the 1- and 12-h fAβ treatments; however, many of the changes seen after oAβ treatment stand in stark contrast to those seen after both fAβ treatments.

To further examine differences and similarities in transcriptional changes between fAβ and oAβ treatments, we directly compared gene expression at 12 h of fAβ treatment (numerator) against gene expression at 12 h of oAβ treatment (denominator) to identify DEGs in these conditions. This comparison revealed disparate changes in transcriptional responses between the conformations of Aβ peptide and identified 982 up-regulated genes and 1,348 down-regulated genes in fAβ versus oAβ treatments (Fig 1D and Table S16). Affected down-regulated genes (down in fAβ, up in oAβ) primarily affected cell cycle and DNA-binding activities (i.e., DNA replication origin binding, kinetochore binding; Fig 1E and Tables S18 and S20), whereas up-regulated genes (up in fAβ relative to oAβ) were enriched in immune system responses (i.e., TAP binding, T-cell receptor binding; Fig 1F and Tables S17 and S19).

### Primary microglia have unique transcriptional responses to Aβ conformations

To directly identify disparate changes in transcription in response to Aβ conformation, we compared the log-fold changes for DEGs in these Aβ treatments (Fig S2). We find a strong correlation (R = 0.74) when comparing treatments of fAβ, 12-h (versus control) against

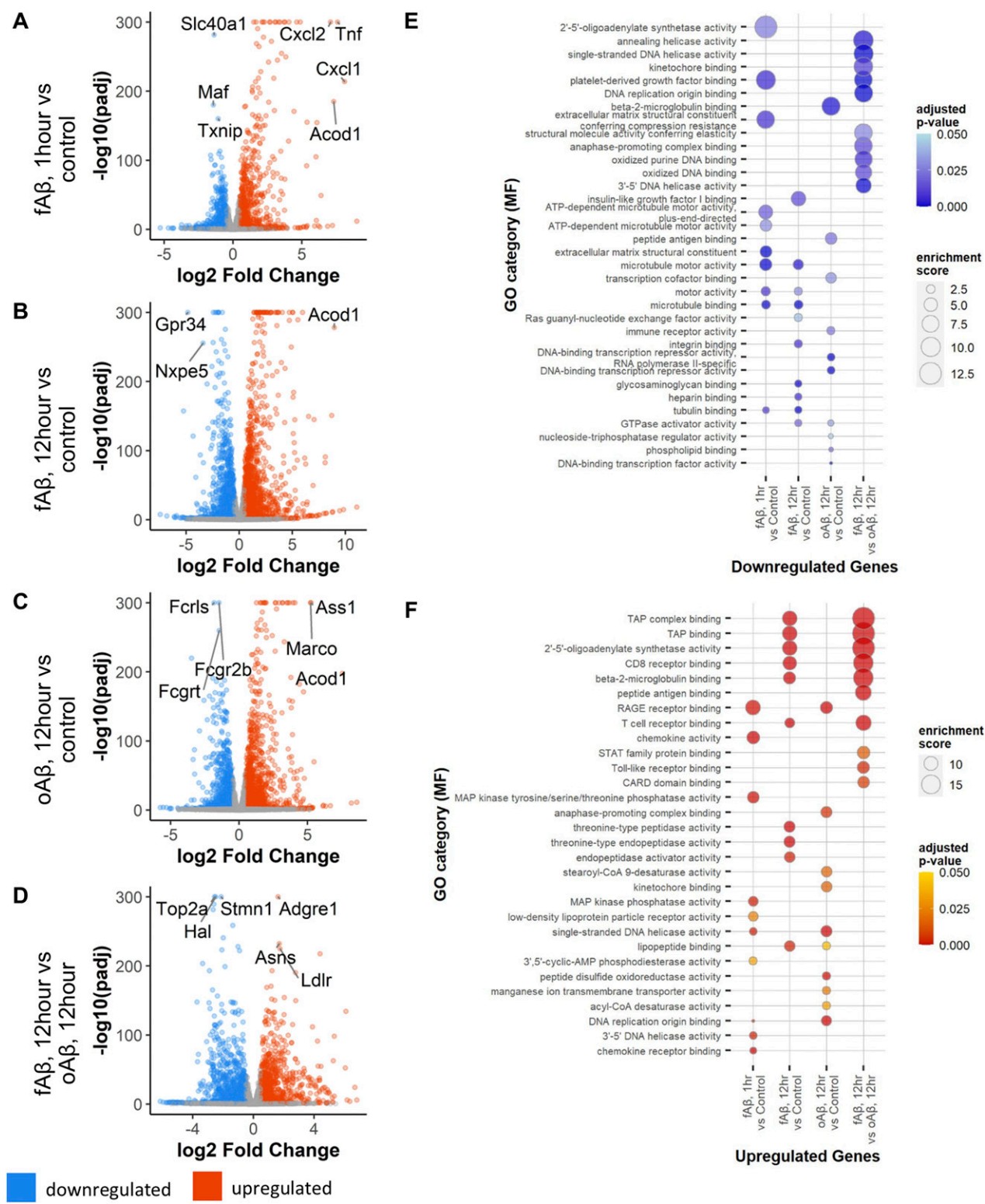

**Figure 1. Differential gene expression in primary microglia after treatment with Aβ42 oligomers (oAβ) or fibrils (fAβ).**
**(A)** Total changes in down- (blue) and up- (red) regulated genes in primary microglia after 1 h of fAβ treatment versus control. **(B)** Volcano plot of differentially expressed genes (DEGs) after 12 h of fAβ42 treatment versus control in primary microglial cultures. **(C)** Volcano plot of DEGs after 12 h of oAβ42 treatment in primary microglial cultures. **(D)** Volcano plot of DEGs after 12 h of Aβ42 fibrils treatment in primary microglial cultures. **(E)** Bubble plots of Gene Ontology (GO) category enrichment results for down-regulated genes. **(F)** Bubble plots of GO category enrichment results for up-regulated genes. Plots for GO category over-enrichment analyses show the top 10 hits for each comparison by enrichment score after a filter step by a P-value adjusted for multiple comparisons of ≤0.05 and keeping GO categories with >5 genes within the category.

oAβ, 12-h (versus control). The 865 commonly up-regulated genes are enriched with GO terms involved with peptidase and chemokine activity (i.e., threonine-type endopeptidase activity), whereas the 865 commonly down-regulated genes are enriched in terms involving posttranslational modifications (histone demethylase activity and ubiquitin-like protein ligase activity) (Fig S2A and B and Tables S21 and S22). Interestingly, the 170 genes that are up-regulated in oAβ, 12-h treatment but down-regulated in fAβ, 12-h treatments are involved in cell cycle (i.e., anaphase-promoting complex) and microtubule motor activities (i.e., motor activity and ATP-dependent microtubule motor activity). The 51 genes down-regulated in oAβ, 12-h treatment but up-regulated in fAβ, 12-h treatment which are involved in antigen binding and immune responses (i.e., TAP complex binding and CD8 receptor binding).

An analysis comparing fAβ, 1-h treatment with oAβ, 12-h treatment reveals similar results (Fig S2C and D and Tables S23 and S24). Commonly up-regulated genes (515 genes) have roles involving the immune system (RAGE receptor binding and chemokine activity) and kinase activities (MAP kinase tyrosine/threonine phosphatase activity), whereas there was no significant enrichment of GO terms (P-value adjusted for multiple comparisons ≤ 0.1) for the 387 commonly down-regulated genes. The divergently responding 56 genes that are up-regulated in oAβ, 12-h treatment but down-regulated in fAβ, 1-h treatment are involved in the cell cycle (anaphase-promoting complex binding) and microtubule motor processes (microtubule motor activity), whereas the 78 genes up-regulated in fAβ, 1-h treatment, but down-regulated in fAβ, 12-h treatment are involved in the innate immune response (complement component C1q complex binding). This analysis highlights the up-regulation of genes involved in the cell cycle and microtubule motor pathways after oAβ treatment.

A direct comparison of significant changes in gene expression between acute 1-h versus longer term 12-h fAβ treatments expose 515 commonly up-regulated genes involved in cytokine and immune activity (immunoglobulin receptor binding and chemokine receptor binding) and 507 commonly down-regulated genes involved in microtubule motor activity (microtubule binding and microtubule motor activity; Fig S2E and F and Tables S25 and S26). Longer term fAβ treatment resulted in up-regulation of 93 genes that are initially down-regulated in 1-h fAβ treatment that are involved in adenylation and GTPase activities (adenylyltransferase activity, GTPase activity, and nucleoside-triphosphatase activity). Acute fAβ, 1-h treatment triggered an up-regulation of 98 genes that are down-regulated after 12 h of treatment which are enriched in diverse GO terms, including complement component C1q complex binding, DNA helicase activity, and integrin binding.

For a more comprehensive view of these disparate changes in microglia after the application of different Aβ species, we examined all genes that were identified as a DEG in any of the of the three treatment paradigms versus control and plotted a heat map of their z-scores with hierarchical clustering of the genes (Fig 2). Clear patterns of transcriptional changes can be seen between conditions. To identify the genes within these clusters, we cut the hierarchical tree at a height of 5.75 which resulted in 13 gene clusters that were then analyzed by GO analysis (Fig 2 and Tables 1 and

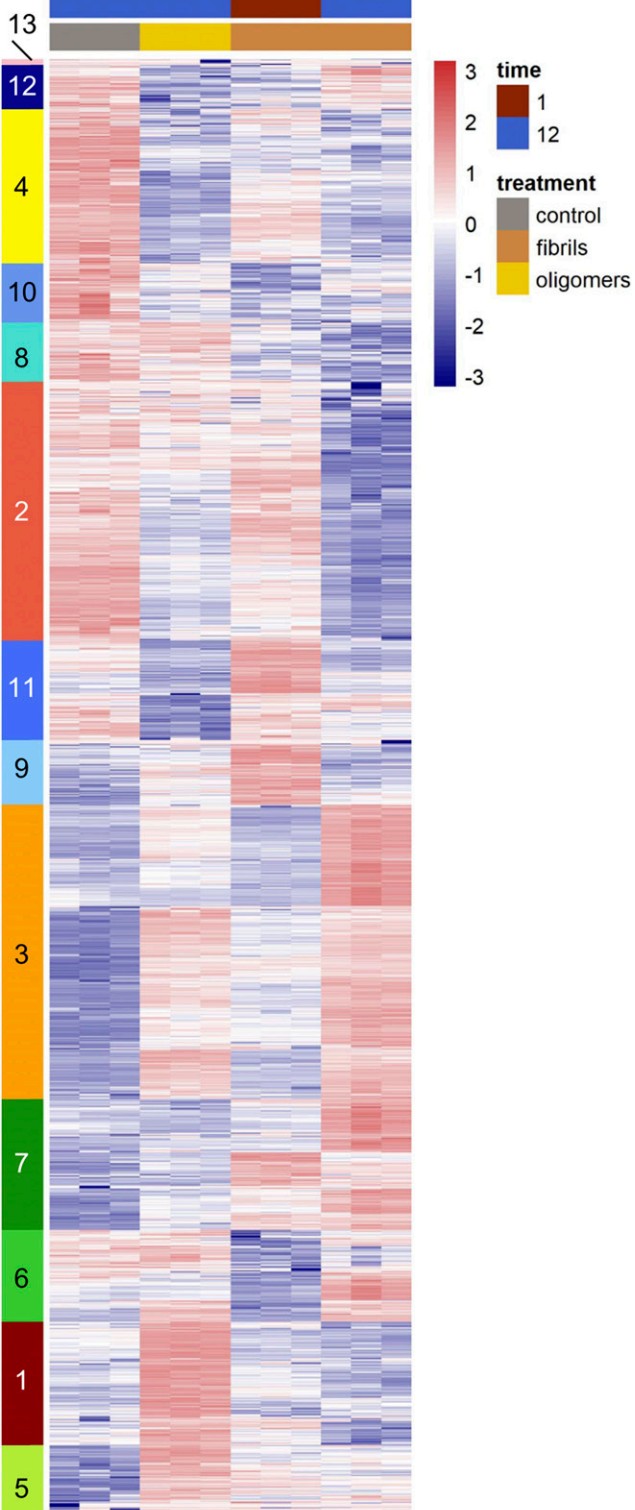

**Figure 2. Hierarchical clustering of differentially expressed genes in Aβ42-treated microglia reveals unique gene signatures.**
Hierarchical clustering Z-scores of gene expression data. A cut height of h = 5.75 was applied to identify clusters of genes with similar expression patterns which produced 13 clusters.

**Table 1. Gene ontology (GO) analysis of differentially expressed gene clusters in Aβ-treated microglia.**

| Cluster | Control | oAβ, 12-h | fAβ, 1-h | fAβ, 12-h | GO category (molecular function) | Enrichment score | *P*-value |
|---|---|---|---|---|---|---|---|
| 13 | ↑ | Mix | Mix | Mix | Carbohydrate:proton symporter activity | 24.95 | $9.15 \times 10^{-3}$ |
| | | | | | JUN kinase activity | 22.46 | $1.02 \times 10^{-2}$ |
| | | | | | Neurexin family protein binding | 17.28 | $1.32 \times 10^{-2}$ |
| 12 | ↑ | ↓ | ↓ | ↑ | Kainate selective glutamate receptor activity | 5.44 | $4.14 \times 10^{-2}$ |
| | | | | | Testosterone dehydrogenase (NAD+) activity | 5.44 | $4.24 \times 10^{-2}$ |
| | | | | | Alpha-adrenergic receptor activity | 5.44 | $4.26 \times 10^{-2}$ |
| 4 | ↑ | ↓ | ↑ | ↓ | ATPase inhibitor activity | 4.66 | $3.16 \times 10^{-4}$ |
| | | | | | RNA polymerase II transcription cofactor binding | 3.11 | $9.28 \times 10^{-3}$ |
| | | | | | LBD domain binding | 2.66 | $1.26 \times 10^{-2}$ |
| 10 | ↑ | ↓ | ↓ | ↓ | Histone demethylase activity (H3-K9 specific) | 8.02 | $3.99 \times 10^{-6}$ |
| | | | | | Leucine binding | 8.02 | $1.64 \times 10^{-3}$ |
| | | | | | Insulin binding | 8.02 | $2.14 \times 10^{-3}$ |
| 8 | ↑ | ↑ | ↓ | ↓ | DNA insertion or deletion binding | 8.09 | $1.49 \times 10^{-3}$ |
| | | | | | MutLalpha complex binding | 8.09 | $1.52 \times 10^{-3}$ |
| | | | | | sphingosine N-acyltransferase activity | 6.94 | $2.06 \times 10^{-3}$ |
| 2 | ↑ | ↓ | ↑ | ↓ | 3-hydroxyacyl-CoA dehydrogenase activity | 2.08 | $4.12 \times 10^{-3}$ |
| | | | | | Co-receptor binding | 1.85 | $1.52 \times 10^{-3}$ |
| | | | | | Dolichyl-phosphate-mannose-protein mannosyltransferase activity | 1.85 | $6.07 \times 10^{-3}$ |
| 11 | ~ | ↓ | ↑ | ↓ | STAT family protein binding | 1.39 | $7.17 \times 10^{-3}$ |
| | | | | | Complement component C1q binding | 1.39 | $7.36 \times 10^{-3}$ |
| | | | | | TAP complex binding | 1.24 | $9.57 \times 10^{-3}$ |
| 9 | ↓ | ~ | ↑ | ↓ | Stearoyl-CoA 9-desaturase activity | 7.39 | $1.72 \times 10^{-3}$ |
| | | | | | MAP kinase tyrosine/serine/threonine phosphatase activity | 6.83 | $8.84 \times 10^{-6}$ |
| | | | | | Acyl-CoA desaturase activity | 6.34 | $2.39 \times 10^{-3}$ |
| 3 | ↓ | ↑ | ↓ | ↑ | Threonine-type endopeptidase activity | 3.25 | $4.46 \times 10^{-14}$ |
| | | | | | Threonine-type peptidase activity | 3.25 | $4.46 \times 10^{-14}$ |
| | | | | | Proteasome-activating ATPase activity | 3.25 | $7.92 \times 10^{-5}$ |
| 7 | ↓ | ↓ | mix | ↑ | TAP binding | 6.99 | $7.76 \times 10^{-10}$ |
| | | | | | TAP complex binding | 6.10 | $6.10 \times 10^{-7}$ |
| | | | | | CD8 receptor binding | 5.99 | $4.93 \times 10^{-8}$ |
| 6 | mix | mix | ↓ | ↑ | Platelet-derived growth factor binding | 6.51 | $6.36 \times 10^{-7}$ |
| | | | | | Extracellular matrix structural constituent conferring compression resistance | 5.58 | $1.57 \times 10^{-6}$ |
| | | | | | Cobalt ion binding | 4.69 | $4.11 \times 10^{-4}$ |
| 1 | ~ | ↑ | ↓ | ↓ | Single-stranded DNA-dependent ATPase activity | 7.97 | $3.77 \times 10^{-18}$ |
| | | | | | Kinetochore binding | 7.76 | $3.05 \times 10^{-6}$ |
| | | | | | Single-stranded DNA-dependent ATP-dependent DNA helicase activity | 7.28 | $2.24 \times 10^{-7}$ |
| 5 | ↓ | ↑ | ~ | ~ | G protein–coupled adenosine receptor activity | 5.31 | $3.35 \times 10^{-3}$ |
| | | | | | Structural constituent of presynapse | 4.13 | $5.78 \times 10^{-3}$ |
| | | | | | Low-density lipoprotein particle receptor activity | 3.71 | $6.96 \times 10^{-3}$ |

GOseq analysis to analyze GO category over-enrichment was applied to these clusters identified in Fig 2. The top three categories are shown after ranking by enrichment score and filtering for genes with a *P*-value of <0.05 and to remove categories with <5 genes within the category.

S27–S29). Cutting the hierarchical tree at this height identified clusters of gene that were visually apparent. By this analysis, we identified clusters of genes that have similarities in expression patterns after treatment with different Aβ conformations. For example, genes in cluster 10 are involved in transcriptional processes and have decreased expression in all three conditions compared with controls. However, this analysis also highlights the clusters of genes that have a unique transcriptional signature in response to specific Aβ conformations. Genes within clusters 11 and 9 have increased expression levels after acute fAβ treatment and are enriched in terms involving metabolic processes and immune responses and cell signaling. Genes in cluster 7 are increased after long-term fAβ treatment and encompass functions of the antigen processing and the immune system. Genes in clusters 1 and 5 are strongly increased in expression after oAβ treatment and are involved in cell cycle and nucleobase metabolism.

## Gene network changes in microglia highlight specific transcriptional responses to Aβ conformations

We applied a weighted gene co-expression network analysis (WGCNA) onto the expression data from Aβ-treated primary microglial cultures. WGCNA is a method to study biological networks by analyzing pair-wise correlations between the genes within the dataset (Langfelder & Horvath, 2008). We identified 71 co-expression modules (Fig 3 and Tables S30 and S33). We correlated the modules with treatment paradigms (Fig 3A) and annotated these modules using a gene overlap analysis (Shen, 2020) with genes identified with subpopulations of microglial cells identified in prior bulk, single-cell (sc-), single-nuclear (sn-) RNA-seq, or spatial transcriptomic studies (Fig 3B and Table S34). We additionally annotated the modules by KEGG and GO analysis to identify enrichment of pathways within the modules (Fig 3C and Tables S31 and S32). By relating the modules to each treatment condition, we observed interesting patterns in module behavior.

Of these modules, 17 modules are positively correlated with all forms of treatment and indicate a nonspecific response to Aβ treatment. These modules include antiquewhite4, brown, coral1, darkseagreen4, honeydew1, lavenderblush3, lightcoral, lightcyan, lightcyan1, lightgreen, lightsteelblue1, orangered3, orangered4, saddlebrown, violet, white, and yellow4. GO and KEGG pathway analysis reveals that genes within these modules are involved in a variety of molecular functions previously linked with AD including cytokine and chemokine activities (lightgreen) the proteosome (brown and saddlebrown), the splicesome (lavenderblush3 and saddlebrown), and neurodegenerative pathways, including AD, Parkinson's disease, and Huntington's disease (brown and saddlebrown). Fifteen modules are negatively correlated with all forms of Aβ treatment—again, indicating a nonspecific response—and include the black, brown4, darkmagenta, darkolivegreen, floralwhite, greenyellow, magenta, mediumpurple3, midnightblue, navajowhite2, paleturquoise, sienna3, skyblue, skyblue2, steelblue, and yellowgreen modules. The genes within these modules are enriched in genes involved in Rab and Ras GTPase activities (mediumpurple3) and with fatty acid metabolism (darkolivegreen).

Six modules are positively correlated with acute, 1-h fAβ treatment and are either negatively correlated or not significantly correlated with the other treatments. These modules characterize the acute response to fAβ treatment and include the tan, salmon, skyblue3, maroon, plum2, and cyan modules. These modules represent genes with functions involved with ion channel activities (cyan), histone modification activity (plum2), RNA processing and splicing, and protein ubiquitination and acetylation (skyblue3).

There are five modules that are positively correlated to long-term, 12-h fAβ and include the darkslateblue, lightpink4, palevioletred3, blue, and coral2 modules. Genes within these modules are enriched with genes with immune/inflammatory/cytokine functions (blue), RNA binding (darkslateblue), GTPase activity (palevioletred and coral2), and transcriptional regulation (lightpink4). In addition, it is within the blue module that most reactive and responsive microglial markers reside (Fig 3C).

Five other modules are positively correlated with long-term, 12-h oAβ treatment and include bisque4, thistle2, mediumorchid, turquoise, and salmon4. These modules are enriched with genes which are involved with extracellular matrix structural components (bisque4) and DNA replication and repair and the cell cycle (turquoise). These analyses further support our original observation that indicates unique microglia transcriptional responses to different species of Aβ peptides.

Interestingly, subpopulations of microglia previously identified in sc-, sn-RNA-seq, or spatial transcriptomic studies did not fall within any single module (Fig 3B). For example, plaque-induced genes (PIGs) which are found in microglia surrounding Aβ plaques (Chen et al, 2020) fall across multiple modules and those modules do not fit any pattern of being correlated or not with any treatment paradigm, including being both negatively and positively correlated with various treatments. This pattern also holds for genes found within the neurodegenerative disease-associated phagocytic microglia cells (DAMs) (Keren-Shaul et al, 2017) and microglia associated with a neurodegenerative phenotype (MGnD) (Krasemann et al, 2017). As noted in these prior studies, the microglia subpopulations share a number of genes in common.

To examine the strength of gene–gene connections with these networks, we chose representative modules that were positively correlated in only one treatment type and examined the networks across all treatments (Fig 4). We plotted edge weights to represent gene–gene connection strengths in an ordered heat map to visualize the overall network strength more easily between the various treatment paradigms. Given that the salmon module has the strongest correlation value with the 1-h fAβ treatment, we used this as a representative network for acute, 1-h fAβ treatment. We find the genes within the salmon module have a stronger overall connection as compared with long-term, 12-h fAβ, or oAβ treatments (Fig 4A). Similar enhancements in gene–gene network strength were seen for the blue module which is positively correlated with long-term, 12-h fAβ treatment (Fig 4B). Although this module does not have the strongest correlation value of the five modules highly correlated with the 12-h fAβ treatments, we chose to examine this module as its member genes are enriched in interferon and immune signaling pathways. Finally, the turquoise

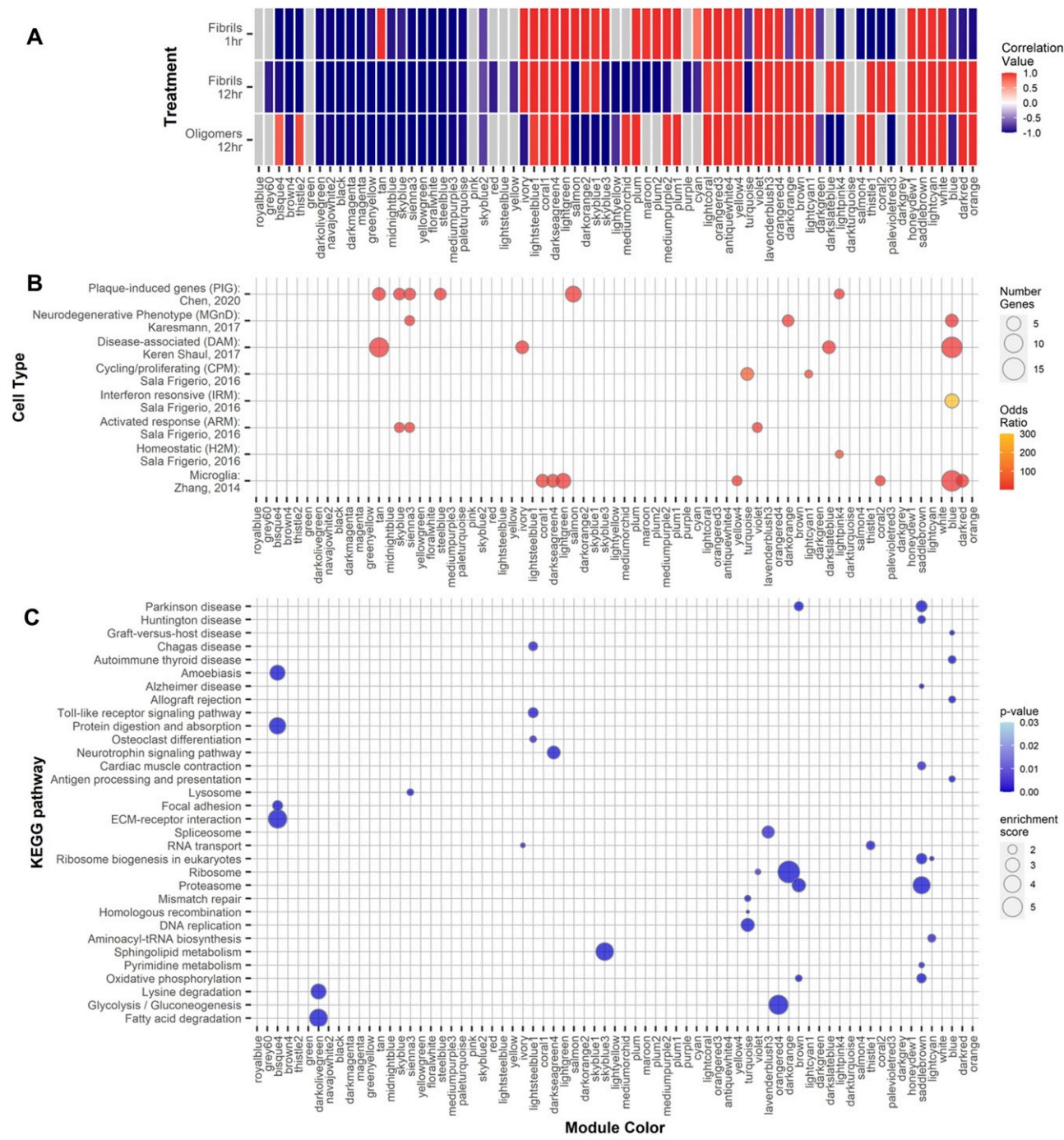

**Figure 3. Weighted gene correlation network analysis.**
Gene modules found by weighted gene co-expression network analysis in Aβ-treated primary microglia. **(A)** Modules are colored in a heat map by their correlation value with the different Aβ treatments. Modules with nonsignificant *P*-values associated or with an absolute correlation value or <0.5 are indicated in gray. **(B)** Bubble plot of a gene overlap analysis to identify shared genes between the module and previously identified microglial subtypes. Modules with significant (*P* ≤ 0.05) odds ratios of overlapping genes are colored as in the scale to the right. The number of overlapping genes is indicated by the dot size. **(C)** KEGG pathway over-enrichment analysis for genes within each module. Pathways with an over-represent *P*-value ≤ 0.05, the number of module genes within the pathway >5, and an enrichment score >1.5 are depicted. *P*-value is indicated by the color scare and the enrichment score by the dot size.

module, which has the highest correlation value of the five modules highly correlated with oAβ treatment, was chosen as the representative module for positive correlation with long-term, 12-h oAβ treatment (Fig 4C). This module also shows a striking increase in network connection strength as compared with the other two conditions. Genes within this module show enrichment in cell cycle, DNA replication, and repair pathways. The top hub genes for these three modules are listed in Table 2.

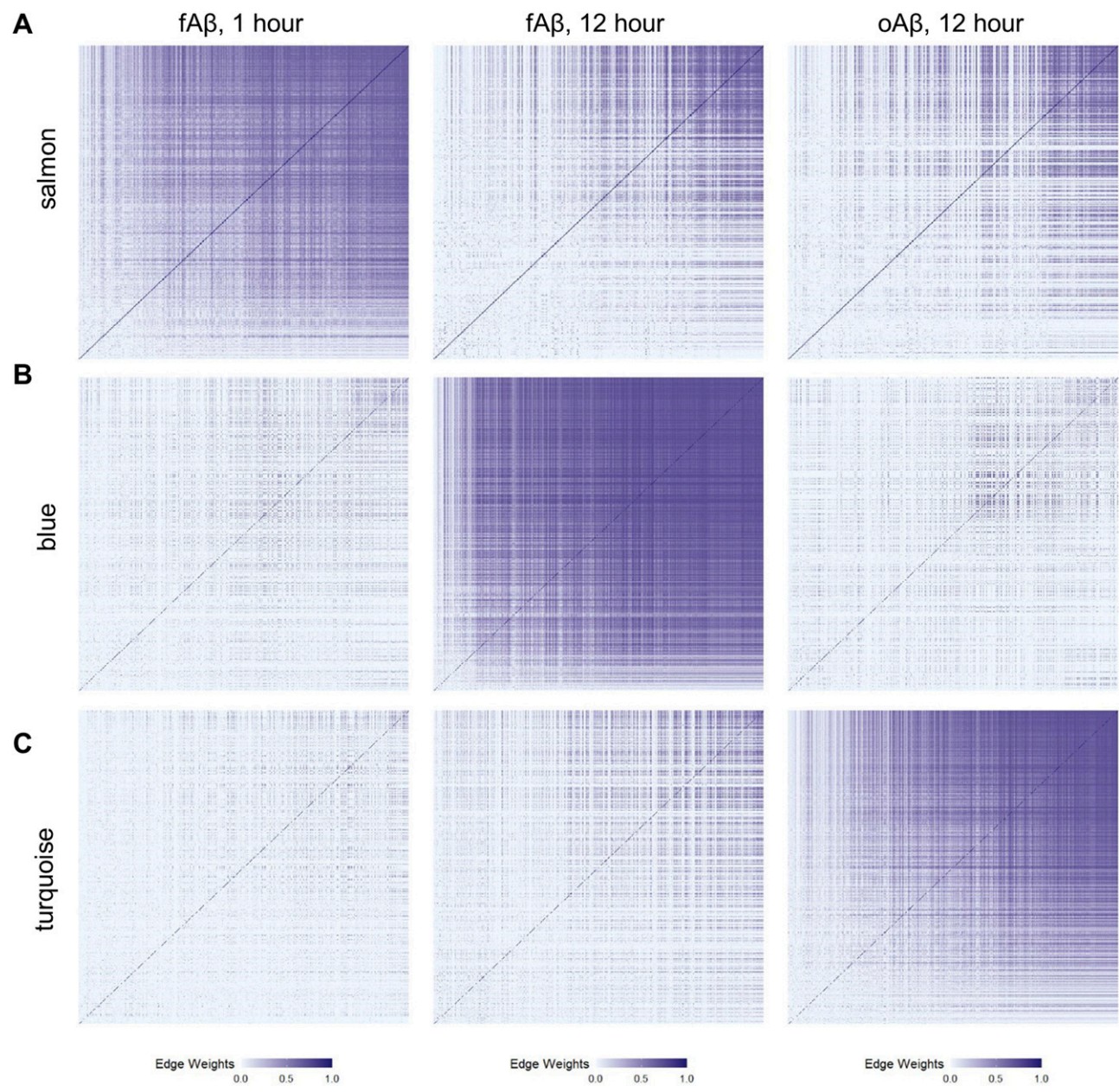

**Figure 4. Gene networks are strongest in the modules that are positively correlated with Aβ treatments.**
Gene networks are shown as heat maps of the edge weight. A greater edge weight (darker blue shades) indicates a strong gene–gene connection. The order of genes within each heat map is preserved for the comparisons across Aβ treatment types. **(A)** The gene network for the salmon module, a representative module positively correlated with acute, 1-h fAβ treatment, is strongest than in 12-h fAβ or 12-h oAβ treatments. **(B)** The gene network for the blue module, a representative module positively correlated with long-term, 12-h fAβ treatment is stronger than in 1-h fAβ or 12-h oAβ conditions. **(C)** The gene network for the turquoise modules, which represent a module positively correlated with long-term, 12-h oAβ treatment, is strongest in oAβ treatments.

## Transcriptional changes in primary microglia do not mimic those seen in the transgenic CRND8 mouse brain

To understand how well ex vivo changes in primary microglia cultures recapitulate in vivo processes, we examined transcriptional changes in the brains of transgenic amyloid mouse model CRND8 at 3, 6, 12, and 20 mo of age by bulk RNA-seq. Using the same cutoff values to identify DEGs as above, we find that at 3 mo of age there are few transcriptional changes between the

transgenic CRND8 and their nontransgenic littermate controls (11 up-regulated and 4 down-regulated, Fig 5A and Table S35). By 6 mo of age, the number of transcriptional changes increases to 187 up-regulated and 105 down-regulated genes (Fig 5B and Table S40). At 12 mo of age, the number of DEGs is higher than at previous time points and is dominated by changes in up-regulated genes (493 up-regulated genes) over those that are down-regulated (103 down-regulated genes, Fig 5C and Table S45). At 20 mo, more genes continue to be up-regulated (746

**Table 2.  Weighted gene co-expression network analysis module statistics.**

| Hub gene | Gene significance | Gene significance *P*-value | Module membership | Module membership *P*-value | kWithin |
|---|---|---|---|---|---|
| Salmon | to fA$\beta$42, 1-h | | | | |
| Gabbr2 | 0.9984 | $8.34 \times 10^{-14}$ | 0.9854 | $5.17 \times 10^{-9}$ | 167.213 |
| Vegfa | 0.9980 | $2.47 \times 10^{-13}$ | 0.9927 | $1.58 \times 10^{-10}$ | 166.033 |
| Cyth1 | 0.9977 | $4.60 \times 10^{-13}$ | 0.9953 | $1.79 \times 10^{-11}$ | 164.974 |
| Rab7b | 0.9976 | $5.71 \times 10^{-13}$ | 0.9903 | $6.51 \times 10^{-10}$ | 159.731 |
| Tnfrsf21 | 0.9971 | $1.72 \times 10^{-12}$ | 0.9929 | $1.42 \times 10^{-10}$ | 164.391 |
| Gpcpd1 | 0.9965 | $3.83 \times 10^{-12}$ | 0.9920 | $2.47 \times 10^{-10}$ | 157.545 |
| Usp2 | 0.9962 | $6.38 \times 10^{-12}$ | 0.9498 | $2.32 \times 10^{-6}$ | 155.563 |
| Folr2 | 0.9961 | $6.94 \times 10^{-12}$ | 0.9800 | $2.46 \times 10^{-8}$ | 167.461 |
| Picalm | 0.9961 | $7.20 \times 10^{-12}$ | 0.9726 | $1.16 \times 10^{-7}$ | 159.591 |
| Plek | 0.9959 | $8.80 \times 10^{-12}$ | 0.9896 | $9.34 \times 10^{-10}$ | 159.217 |
| Blue | to fA$\beta$42, 12-h | | | | |
| Tmem176a | 0.9997 | $3.13 \times 10^{-17}$ | 0.9513 | $1.98 \times 10^{-6}$ | 468.954 |
| Acp2 | 0.9996 | $4.65 \times 10^{-17}$ | 0.9321 | $1.02 \times 10^{-5}$ | 462.765 |
| Gpr18 | 0.9996 | $5.40 \times 10^{-17}$ | 0.9432 | $4.24 \times 10^{-6}$ | 450.890 |
| Fnbp1l | 0.9995 | $1.77 \times 10^{-16}$ | 0.9736 | $9.61 \times 10^{-8}$ | 452.921 |
| Tmem176b | 0.9995 | $2.50 \times 10^{-16}$ | 0.9316 | $1.05 \times 10^{-5}$ | 479.564 |
| Adgre1 | 0.9994 | $5.13 \times 10^{-16}$ | 0.9676 | $2.65 \times 10^{-7}$ | 485.170 |
| Slc11a2 | 0.9994 | $5.51 \times 10^{-16}$ | 0.9240 | $1.76 \times 10^{-5}$ | 487.036 |
| Cep85 | 0.9991 | $5.73 \times 10^{-15}$ | 0.9440 | $3.94 \times 10^{-6}$ | 463.008 |
| Cd82 | 0.9989 | $1.12 \times 10^{-14}$ | 0.9493 | $2.43 \times 10^{-6}$ | 462.487 |
| Nr1h3 | 0.9989 | $1.34 \times 10^{-14}$ | 0.9719 | $1.33 \times 10^{-7}$ | 466.064 |
| Turquoise | to oA$\beta$42, 12-h | | | | |
| Asf1b | 0.9998 | $6.09 \times 10^{-19}$ | 0.991 | $3.71 \times 10^{-10}$ | 400.289 |
| Alox5 | 0.9998 | $2.49 \times 10^{-18}$ | 0.933 | $9.27 \times 10^{-6}$ | 402.563 |
| S100a4 | 0.9998 | $3.79 \times 10^{-18}$ | 0.973 | $1.18 \times 10^{-7}$ | 400.107 |
| Klf2 | 0.9996 | $4.53 \times 10^{-17}$ | 0.952 | $1.89 \times 10^{-6}$ | 402.789 |
| Hal | 0.9996 | $7.56 \times 10^{-17}$ | 0.868 | $2.49 \times 10^{-4}$ | 401.621 |
| Top2a | 0.9996 | $7.82 \times 10^{-17}$ | 0.996 | $6.24 \times 10^{-12}$ | 401.975 |
| Cks1b | 0.9996 | $8.13 \times 10^{-17}$ | 0.950 | $2.16 \times 10^{-6}$ | 403.304 |
| Pygl | 0.9991 | $3.81 \times 10^{-15}$ | 0.961 | $6.31 \times 10^{-7}$ | 403.646 |
| E2f1 | 0.9991 | $5.78 \times 10^{-15}$ | 0.965 | $3.66 \times 10^{-7}$ | 402.021 |
| Mcm7 | 0.9989 | $1.35 \times 10^{-14}$ | 0.958 | $9.81 \times 10^{-7}$ | 400.930 |

Top 10 hub genes within the modules depicted in Fig 3 identified by weighted gene co-expression network analysis. Results are sorted by gene significance to each treatment type. Module statistics including gene significance value (to treatment), *P*-values corresponding to the gene significance (*P*-value), module membership value (of gene to module), module membership *P*-value, and gene connectivity within the module (kWithin) are shown.

genes) than down-regulated (115 genes, Fig 5D and Table S50). Trends for GO term enrichment in down-regulated genes was not evident until 12 mo with enriched terms including a variety of receptor binding activities (i.e., glucocorticoid receptor binding and steroid hormone receptor activity) and involvement of core promoter activity (core promoter sequence-specific DNA binding; Fig 5E and Tables S37, S39, S42, S44, S47, S49, S52, and S54). Up-regulated genes are enriched primarily with immune responses (immunoglobulin receptor activity and IgG binding) that are

consistent as the mice age (Fig 5F and Tables S36, S38, S41, S43, S46, S48, S51, and S53).

Not surprisingly, direct comparisons of the microglial-specific genes in A$\beta$-treated primary microglia with transgenic CRND8 mice are poorly correlated (Figs 6 and S3). Correlation values are low between differentially expressed microglial genes in transgenic CRND8 mice at 20 mo versus A$\beta$-treated primary microglia for any treatment paradigm, oA$\beta$ 12-h (Fig 6A), fA$\beta$ 12-h (Fig S3A), or fA$\beta$ 1-h (Fig S3B). In the transgenic CRND8, these genes are nearly

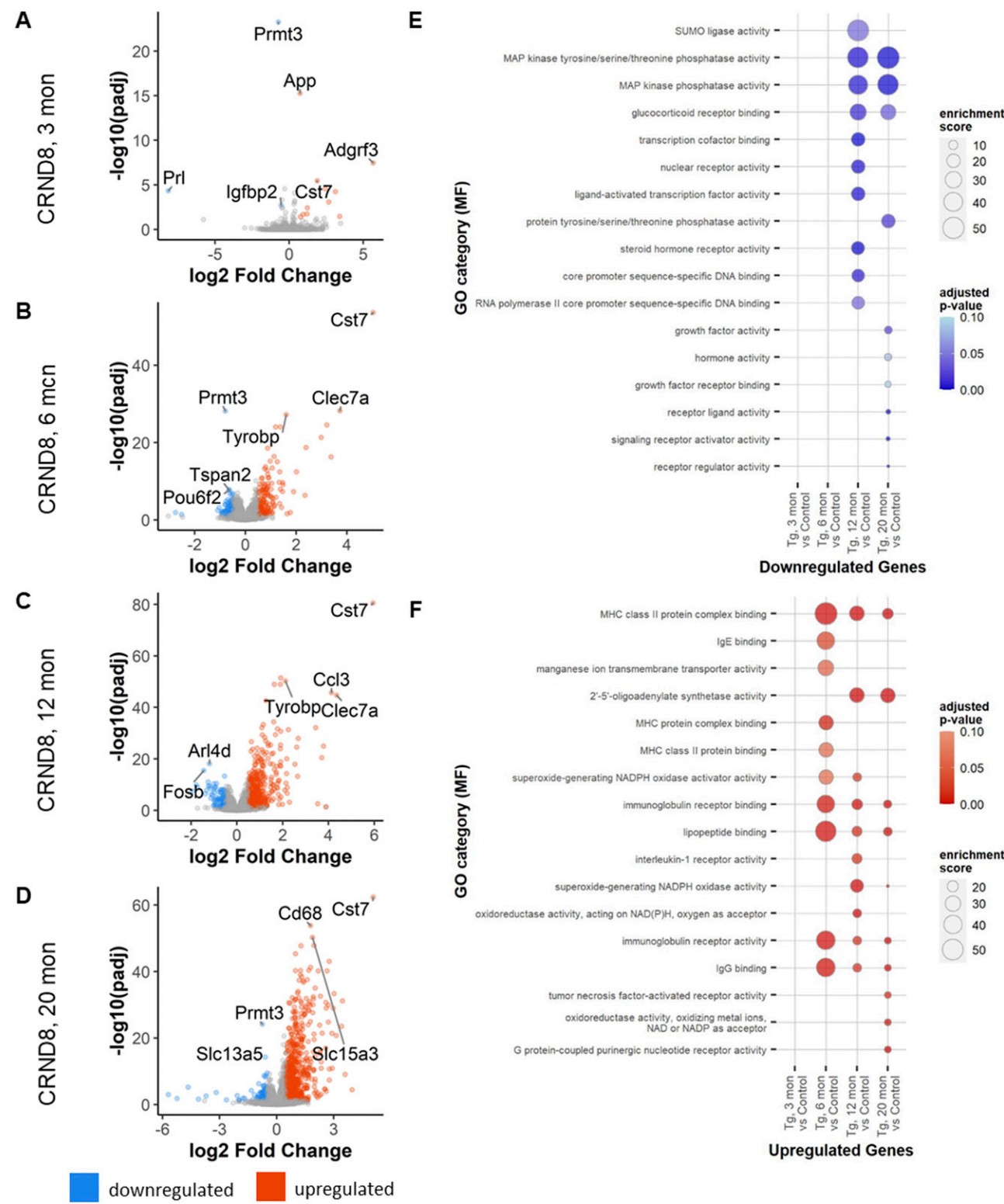

**Figure 5. Differential gene expression in transgenic CRND8 mice.**
**(A)** Total changes in down- (blue) and up- (red) regulated genes in transgenic CRND8 mouse brains versus nontransgenic controls at 3 mo. **(B)** Total changes in down- and up-regulated genes in transgenic CRND8 mouse brains versus nontransgenic controls at 6 mo. **(C)** Total changes in down- and up-regulated genes in transgenic CRND8 mouse brains versus nontransgenic controls at 12 mo. **(D)** Total changes in down- and up-regulated genes in transgenic CRND8 mouse brains versus nontransgenic controls at 20 mo. **(E)** Bubble plots of Gene Ontology (GO) category enrichment results for down-regulated genes. **(F)** Bubble plots of GO category enrichment results for up-regulated genes. Plots for GO category over-enrichment analysis show the top 10 hits for each comparison by enrichment score after a filter step by a *P*-value adjusted for multiple comparisons of ≤0.1 and keeping GO categories with >5 genes within the category.

**Figure 6. Microglia transcriptional responses at the individual gene level are not reflective of changes seen in the CRND8 model. (A, B)** Comparisons of log$_2$ fold change values for microglial genes (Zhang et al, 2014) in transgenic CRND8 versus oAβ treatment in primary microglia show little correlation. Geometric means of FPKM data of representative genes differentially expressed in Ab-treated primary microglia is shown for Aβ-treated microglia (top row) and CRND8 mouse brains (bottom row) (B). **(C, D, E, F)** Similar plots are shown for representative differentially expressed genes identified in CRND8 mice (C), Alzheimer's disease-relevant genes (D), representative cytokine genes (E), and representative cytokine receptor genes (F).

universally up-regulated, but are both up- and down-regulated in the primary microglia. We examined representative genes that are highly differentially expressed in Aβ-treated microglia (Fig 6B) which reveal little (*Vim*) to no (*Sod2, Sgk1*) corresponding changes in the transgenic CRND8 mice over time—indeed some changes were opposite of those observed in CRND8 (Fcgr2b). Conversely, examining a selection of genes that are consistently and significantly changed in transgenic CRND8 mice over time (Fig 6C; *Cst7*, *Irf8*, and *Plek*) exposes variable responses in microglia after the application of either fAβ or oAβ peptide. In addition, a panel of Alzheimer's-disease relevant genes, which are consistently up-

regulated in the transgenic CRND8 mouse brain over time, also reveals variable (*Abi3* versus *Plcg2*)—and sometimes unexpected (*Trem2*)—responses to Aβ peptides in microglia (Fig 6D). This pattern is also seen in a selection of cytokines (Fig 6E; *Ccl3, Ccl4* and *Tnf*) and cytokine receptor (Fig 6F; *Ccr5, Csf3r*, and *Tnfrsf1a*) genes.

We then examined the transcriptional profile of microglial cell subsets that have been identified in past sc-, sn-RNA-seq, or spatial transcriptomic studies of microglia (Fig 7). As evidenced by the increase in the transgenic CRND8 brains, the expression of these genes within the subpopulations is increased in AD. A general

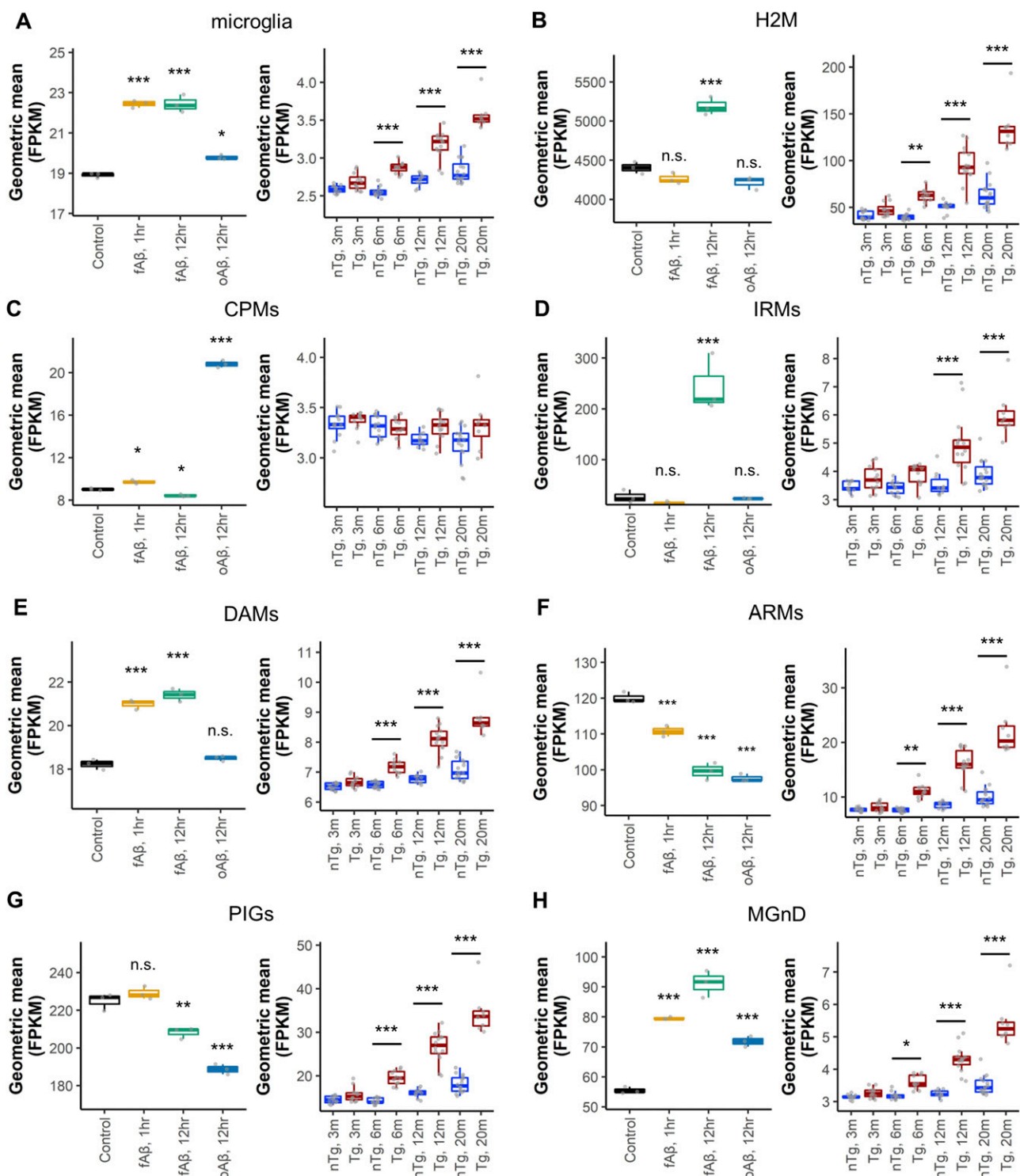

**Figure 7. Microglia subtype transcriptional signatures in primary microglia do not reflect changes seen in the CRND8 model.**
Gene signatures for microglia genes and sub-populations of microglia are shown for primary microglia cultures (left) and CRND8 mouse brains (right). **(A)** Microglia expression signature identified in Zhang et al (2014). **(B)** Activated microglia expression signature in Aβ-treated microglia. **(B)** Homeostatic (H2M) microglial gene expression signature as in Sala Frigerio et al (2019). **(C)** Cycling and proliferating microglia gene expression signature as in Sala Frigerio et al (2019). **(D)** Interferon-responsive microglia gene expression signature as in Sala Frigerio et al (2019). **(E)** Disease-associated microglia gene expression signature identified in Keren-Shaul et al (2017). **(F)** Activated response microglia gene expression signature as in Sala Frigerio et al (2019). **(G)** Plaque-induced microglia gene expression signature as in Chen et al (2020). **(H)** Neurodegenerative microglia phenotype (MGnD) gene expression signature as in Krasemann et al (2017). *P-adj < 0.05; **P-adj < 0.01; ***P-adj < 0.001. FPKM, fragments per kilobase per million mapped fragments.

transcriptomic signature of microglial-enriched genes (Zhang et al, 2014) is increased after all Aβ treatments in microglia primary cultures—a signal that mimics increases seen in transgenic CRND8 mice over time (Fig 7A). A homeostatic microglia (H2M) signature (Sala Frigerio et al, 2019) increases over time in transgenic CRND8 mice, but this increase is seen only in microglial cultures treated for 12-h fAβ (Fig 7B). We examined the transcriptional signature associated with cycling and proliferative microglia (Sala Frigerio et al, 2019) (Fig 7C). There is a large increase in the cycling and proliferating microglia signature in oAβ-treated microglia, but no difference is seen in the transgenic CRND8—which stands as a contrast to the general trend in the other microglial subtypes. This likely reflects that this population represents a very small percentage of microglial cells within the brain (Sala Frigerio et al, 2019), and its signature is lost within the larger milieu of other cell types within the brain. We also examine interferon-responsive microglia (Fig 7D) (Sala Frigerio et al, 2019). This transcriptional signature increased over time in transgenic CRND8 but a large change is seen only in response to long-term fAβ treatment. Interestingly, an increased transcriptional response in the disease associated microglia profile (DAMs, found in the microglia surrounding Aβ plaques [Keren-Shaul et al, 2017]) is seen in response to fAβ, but not oAβ treatment, whereas a steady increase is seen in the transgenic CRND8 (Fig 7E). Intriguingly, transcriptional responses linked to both activated response microglia (Fig 7F [Sala Frigerio et al, 2019], which are responsive to Aβ deposition) and PIGs (Fig 7G [Chen et al, 2020]) are decreased or unchanged in all treatment paradigms in primary microglia while these genes steadily increase over time in transgenic CRND8. In contrast, genes linked to the microglial neurodegenerative phenotype (MGnD, Fig 7H) (Krasemann et al, 2017) appear as a likely reliable indicator of transcriptional changes for all Aβ treatment paradigms as well as in transgenic CRND8 brains.

## Discussion

Acute exposure of cultured primary microglia to oAβ or fAβ elicits a robust and rapid transcriptional response. Both forms of Aβ induce significant increases and decreases in RNA levels for hundreds of genes. Nevertheless, transcriptomic responses to oAβ and fAβ at 12-h are distinguishable. Of note, the finding that oAβ increases RNAs associated primarily with cell cycle, whereas fAβ increases RNAs associated primarily with phagocytic processes is intriguing.

As there are numerous validated and candidate Aβ receptors expressed on microglia (Jarosz-Griffiths et al, 2016), such studies indicate that acute exposure to Aβ aggregates induces robust cellular events that can be assessed at the systems level using transcriptomic approaches. Based on the studies of fAβ there is a clear temporality to the response with varying clusters of genes changing in both similar and different directions at the various time points. These data are reminiscent of studies examining acute effects of LPS on primary microglia, although given numerous experimental differences with historical datasets a much more systematic, side by side, comparison would be needed to evaluate the overall similarity in response to classic proinflammatory mediators such as LPS and Aβ.

As we and others have used primary microglia to study uptake and clearance of Aβ and Aβ aggregates, a primary objective of this study was to determine if the response to Aβ in such acute studies is indicative of system levels changes in mouse models of Aβ deposition, where Aβ accumulates over time. In this case, we have compared the transcriptomic changes in CRND8 transgenic model (compared with nontransgenic controls) with our acute transcriptomic signatures of the primary microglia exposed to Aβ. These data reveal that acute transcriptional responses of primary microglia to Aβ poorly reflect the in vivo responses of genes to chronic progressive Aβ accumulation. Like a number of other recent studies, these data suggest that although primary microglial studies may have utility in some settings, extrapolating results from these studies to the in vivo setting is problematic.

Many laboratories in the field, including our own, have focused on responses of microglia to classic cytokines including but not limited to TNFα, IL1α, IL1β, IL10, IL6, and IFNγ (Chakrabarty et al, 2010, 2011, 2015; Colon-Perez et al, 2019; Webers et al, 2020). Although these cytokines show massive changes in transcript in primary culture, in vivo transcript levels in the brain are very low throughout the lifespan of the non-Tg and Tg mice. Although some cytokines show small increases over time in the presence of amyloid deposition, the magnitude of this increase is nowhere near the scale of increase observed in the primary culture. The massive increases in transcript levels observed in primary microglial cultures of many of these cytokines and other immune factors have likely contributed to the field's focus on these as key mediators of the microglial responses to Aβ and other insults. However, data presented here, as well as other studies (Butovsky et al, 2014), highlight differences between the ex vivo and in vivo microglial responses and indicate that the focus on some of these cytokine and other immune factors may be misleading.

Notably, microglia—at least at the transcript level—express moderate to high levels of many classic cytokine receptors in vivo. Perhaps, the low level of ligand expression compared with relatively high levels of receptor would suggest that these receptors on microglia serve primarily to sense non-CNS changes in the cytokine levels after peripheral insults. In any case, these data along with numerous other studies demonstrating the heterogeneity of microglia in vivo (Butovsky et al, 2014; Keren-Shaul et al, 2017; Krasemann et al, 2017; Friedman et al, 2018; Hammond et al, 2019; Chen et al, 2020; Olah et al, 2020) highlight the notion that primary isolated microglia cells are poor proxies for in vivo responses. As study of microglial cells in the brain has many limitations, additional efforts to develop better ex vivo models of microglial responses would benefit the field. Although several reports of such efforts exist (Arber et al, 2017; Croft et al, 2019), further evaluation and "stress-testing" of these and other ex vivo methods will be needed before they are likely to be widely adopted.

Previous studies have focused on the functional consequences of treating various primary CNS cells with oAβ or fAβ. oAβ species have been conceptualized by some in the field as the proximal neurotoxin in AD (Haass & Selkoe, 2007; Wang et al, 2016; Cline et al, 2018; Li & Selkoe, 2020), as they disrupt synaptic transmission in neurons at very low, picomolar concentrations (Waters, 2010; Rammes et al, 2011). However, the evidence that oAβ species are overtly toxic with respect to inducing neuronal death is lacking;

further there is debate as to whether appreciable concentrations of intrinsically soluble oligomers exists in the AD brain or mouse models of amyloid deposition (Tseng et al, 1999; Jan et al, 2008, 2011; van Helmond et al, 2010). In contrast, at least in primary neuronal cultures, higher concentrations of various aggregates have been linked to induction of neuronal death via apoptotic mechanisms (Deshpande et al, 2006). Both direct toxicity of the aggregates or aggregate growth and indirect toxicity via activation of glial cells that results in neurotoxicity have been invoked as mechanisms underlying Aβ induced neuronal death (Kayed & Lasagna-Reeves, 2013). Clearly, the massive alterations in microglial cells observed here in response to synthetic Aβ aggregates reinforces the potential for neurotoxicity in mixed primary cultures. However, we would note that both oAβ and fAβ induce massive changes in the transcriptome of microglia and certainly lend little credence to claims by some in the field that fAβ is inert (reviewed in Walsh and Selkoe [2007]). Indeed, our results suggest that microglial transcriptional responses to fAβ more closely mimic in vivo responses to amyloid accumulation as evidenced by the behavior of the "blue" module genes in our study which positively correlated with fAβ treatment are paralleled in the transgenic CRND8 brain and the microglial subtype analysis.

As suggested above, the concept that microglial cells might make exquisitely sensitive biosensors that can be used to distinguish between various aggregate forms is intriguing. Microglial do appear at the transcript level to respond in partially overlapping, but distinct ways to oAβ or fAβ. Much more extensive studies will be needed to follow up on this intriguing observation. However, from a teleological point of view this concept makes quite a bit of sense. Microglial cells with a plethora of damage associated and pathogen associated receptors are designed to respond rapidly to potentially harmful proteins and other stimuli (Deshpande et al, 2006). One would predict that overlapping but distinct binding interactions could result in partially overlapping but distinct responses that might essentially provide a type of integration of signals to distinguish various aggregates.

As the main goal of these studies was to assess the system level responses of microglial cells in culture to Aβ aggregates and compare that to a longitudinal transcriptomic study in amyloid precursor protein (APP) mice, there are a number of limitations that are worth noting. First, preparations of oligomeric and fibrillar forms of Aβ were not purified; however, the method of preparation is one that is typically used in many laboratories and each preparation have distinctive features that evident on Western blotting. In addition, these preparations elicit distinct transcriptional responses as outlined in this study. Second, both dose response and more extended time courses were not conducted. Third, we did not include monomeric Aβ42, as it would likely aggregate at these concentrations during incubation; nor did we include a short-term oAβ42 time point. Finally, we have not pursued studies to determine whether fAβ and oAβ induce different functional states in the cultured microglia cells. It is almost certain such studies would yield interesting data, but it is unlikely that it would alter the relevance of the work with respect to disease implications in AD.

A recent elegant study exploring in vivo microglial responses to LPS using translational profiling approaches to assess both ribosome-associated transcripts and proteins showed major discrepancies between the proinflammatory transcriptomic signature and a more immune modulatory and homeostatic protein signature (Boutej et al, 2017). Given the massive up-regulation of proinflammatory transcripts in cultured microglia exposed to Aβ and the large number of up-regulated microglial transcripts in APP mouse models and human AD, it will be important to integrate proteomic and transcriptomic studies of microglia in the future. Indeed, at least in the Boutej study, the biologic inferences derived from evaluating the proteome or transcriptome are disparate and only when the two are compared directly does the concept of widespread translation repression emerge. Additional studies also show that even the process of rapid isolation of microglial cells from the brain changes their transcriptome (Tham et al, 2003; Lin et al, 2017; He et al, 2018). Thus, even though single-cell transcriptomic and proteomic studies of isolated microglia cells potentially provide new insights into their roles in health and disease, additional validation using in situ methodologies is needed to confirm that changes observed reflect changes in situ and are not induced during the isolation.

The number of studies focusing on microglia cells and their impact on AD and other neurodegenerative disorders is rapidly expanding. This study and many others highlight that traditional methods to study them, such as in primary cultures, are highly artificial and may lead to inappropriate conclusions. Current efforts to develop strategies to harness microglial function in a therapeutically beneficial fashion must by necessity study the effect of that therapy in vivo. However, given the large number of immune factors that are emerging as modulators of neurodegenerative pathologies, and the limitations of only studying these cells in vivo, additional efforts to validate ex vivo systems that better approximate microglial functions in vivo ware warranted.

# Materials and Methods

### Animal research

All animal research was performed under protocols approved by the Institute for Animal Care and Use Committee at the University of Florida.

### Microglial primary cultures and Aβ42 treatment

Mouse pups for primary microglial cultures are obtained from matings of B6/C3HF1 mice (Envigo). Mice are given ad libitum access to food and water and are maintained on a 12-h light/12-h dark cycle. Primary microglia cultures were isolated following described protocols (Rosario et al, 2016). Briefly, cortices were isolated at postnatal day P2-P3. The mixed microglial/astrocyte cultures were maintained in 75-cm$^2$ flasks with 20 ml of DMEM containing 10% fetal bovine serum. After 10 d, the flasks were shaken on a rotary platform for 30 min at 37°C at 150 rpm to dislodge the microglia from the adherent astrocyte layer. The microglia were plated into six-well plates and maintained at 37°C. 1 d after plating, microglia were treated with 5 µM Aβ$_{42}$ fibrils or oligomers for 1- or 12 h as

noted. Cells were washed with PBS before harvest. Three replicates for each condition were performed.

## Fibrillar and oligomeric Aβ preparation

Fibrillar and oligomeric forms of Aβ were prepared as previously described (Stine et al, 2003; Chakrabarty et al, 2018). Aliquots (10, 100, and 1,000 ng) were separated on SDS–PAGE page run using Biorad Criterion 10% bis-tris gel and XT running buffer/sample buffer for 60 m at 180 V (constant). Gel transferred onto 0.2-micron PVDF in Towbin transfer buffer for 45 m at 150 V (constant). 6E10 (BioLegend) primary antibody diluted at 1:1,000 and applied for 1.5 h at 37°. Primary antibody was detected with goat anti-mouse IR700 and scanned on LiCor Odyssey 700 mm channel.

## RNA extraction and sequencing

Microglial RNA was extracted using the RNeasy mini extraction kit with on-column DNase treatment (QIAGEN). RNA quality was determined with the Qubit RNA HS assay. RNA quality was checked via RIN on an Agilent Bioanalyzer 2100 with the Eukaryote Total RNA Nano chip. Libraries were generated with the Illumina RNA-seq library prep for low input RNA. Libraries were sequenced on paired-end, 75 bp runs on the Nextseq 500 (Illumina). RNA QC, library preparation, and sequencing were performed at the University of Florida's Interdisciplinary Center for Biotechnology Research sequencing core.

## Transgenic CRND8 RNA-sequencing data

Data for the transgenic CRND8 mice were obtained from the AMP-AD Knowledge Portal (doi: 10.7303/syn3157182). Experimental details are located within the data portal's Web site. BAM files were downloaded from the AD Knowledge portal and used with the analysis method described below. Animal numbers are as follows: 3-mo, nTg-F: 6; 3-mo, nTg-M: 6; 3-mo, Tg-F: 6; 3-mo, Tg-M: 6; 6-mo, nTg-F: 5; 6-mo, nTg-M: 7; 6-mo, Tg-F: 5; 6-mo, Tg-M: 6; 12-mo, nTg-F: 5; 12-mo, nTg-M: 5; 12-mo, Tg-F: 7; 12-mo, Tg-M: 7; 20-mo, nTg-F:11; 20-mo, nTg-M: 5; 20-mo, Tg-F: 5; and 20-mo, Tg-M: 3. Male and female mice of the same age and genotype were grouped together for this analysis.

## RNA-seq analysis

### FASTQ alignment, gene counts, and differential expression analysis

Resulting FASTQ files were aligned against the mouse genome (GRCm38) and GRCm38.94 annotation using STAR v2.6.1a (Dobin et al, 2013) to generate BAM files. BAM files were used to generate gene counts were generated using Rsamtools (Morgan et al, 2018) and the summarizeOverlaps function with the GenomicAlignments package (Lawrence et al, 2013). Differential gene expression analysis was performed with DESeq2 package using the "DESeq" function with default settings (Love et al, 2014) which fits a generalized linear model for each gene. Subsequent Wald test *P*-values are adjusted for multiple comparisons using the Benjamini–Hochberg method (adjusted *P*-value). Pair-wise changes in gene expression levels were examined between groups to identify DEGs. DEGs were defined as an absolute log$_2$ fold change ≥0.5 and an adjusted *P*-value ≤0.05.

## WGCNA

The WGCNA package in R (Langfelder & Horvath, 2008, 2012) was used to construct gene correlation networks from the expression data after filtering and removing genes with zero variance. For the microglia dataset, a soft power setting of nine was chosen using the "pickSoftThreshold" function within the WGCNA package. The network was constructed using all microglial samples. Adjacency matrices were constructed using expression data and this power setting with the "adjacency" function and a signed hybrid network. Module identification was performed using the "cutreeDynamic" function and a deepSplit setting of two with a minimum module size of 30 for all analyses.

## Functional annotation of DEGs, heat map clusters, and WGCNA modules

Gene ontology enrichment analysis was performed with goseq v1.42.0 (Young et al, 2010) to identify gene ontology categories—focusing on the molecular function category—and KEGG pathways that are affected between the various conditions. For DEGs, up- and down-regulated gene lists were analyzed separately. For WGCNA, gene lists from each module were used as input. Over-represented *P*-values were adjusted for multiple comparisons using the Benjamini–Hochberg adjustments for controlling false-discovery rates. An enrichment score was calculated by an observed-over-expected ratio of

$$(DEG/totalDEG)/(CategoryTotal/GeneTotal),$$

where *DEG* represents the total number of DEGs or module genes within the GO or KEGG category, *totalDEG* represents the total number of DEGs or module genes; *CategoryTotal* represents the total number of genes within the GO or KEGG category, and *GeneTotal* represents the total number of genes examined. GO terms and KEGG pathways are filtered for *P*-values adjusted for multiple comparisons (BHadjust) <0.05 (Aβ-treated microglia) or 0.1 (CRND8 mice), enrichment scores >1, and total number of genes within the category >5.

Z-scores for genes identified as a DEG for any Aβ-treatment comparison versus control were plotted in a heat map using pheatmap v1.0.12. Clusters were identified using the cutree function with h = 5.75. goseq was used for GO and KEGG pathway analysis on genes within each cluster. GO terms and KEGG pathways are filtered for *P*-values < 0.05, enrichment scores >1, and total number of genes within the category >5.

Gene lists to annotate WGCNA modules and identify microglia subtype signatures were identified from previously published studies (Zhang et al, 2014; Keren-Shaul et al, 2017; Krasemann et al, 2017; Friedman et al, 2018; Hammond et al, 2019; Sala Frigerio et al, 2019; Chen et al, 2020) (see also Table S33). Gene overlap analysis was conducted with the GeneOverlap package in R (Shen, 2020). GeneOverlap uses Fisher's exact test to calculate the *P*-value for significance testing and calculating the odds ratio. goseq was used for GO and KEGG pathway analysis of genes within each module filtering for those terms with *P*-values < 0.05, enrichment scores >1, and total number of genes within the category >5.

## Direct comparisons of DEGs between treatment types

DEG datasets for each treatment paradigm against control were filtered for significant gene changes using criteria described above. The distribution of resulting $\log_2$ fold change was tested for a normal distribution using the Shapiro–Wilk normality test. The correlation value for the $\log_2$ fold change value in each pair-wise comparison was calculated using Spearman's rank-order correlation test at a confidence level set to 0.95 in R and graphs were drawn using the ggpubr package in R (Kassambara, 2020).

## Statistical analysis and data visualizations

ANOVA with Tukey's post hoc multiple comparisons test was performed in R. Data visualizations were generated in R using the ggplot2 package (Wickham, 2016) unless otherwise noted. Bar plots show mean ± SD. For boxplots, upper, middle, and lower hinges correspond to first quartile, median, and third quartiles, respectively. Upper (or lower) whiskers correspond to the largest (or smallest) observation beyond the upper hinge up to 1.5 times the interquartile range. Outliers beyond the upper and lower whiskers are plotted independently.

# Data Availability

FASTQ files for the Aβ-treated primary microglia samples are available via the AD Knowledge Portal (https://adknowledgeportal.org). The AD Knowledge Portal is a platform for accessing data, analyses, and tools generated by the Accelerating Medicines Partnership (AMP-AD) Target Discovery Program and other National Institute on Aging-supported programs to enable open-science practices and accelerate translational learning. The data, analyses, and tools are shared early in the research cycle without a publication embargo on secondary use. Data are available for general research use according to the following requirements for data access and data attribution (https://adknowledgeportal.org/DataAccess/Instructions).

For access to content described in this manuscript, see: http://doi.org/10.7303/syn25006578. Interactive data portals are available for viewing at the following: Aβ-treated microglial DEG data: https://tinyurl.com/y3q3kaoe. CRND8 DEG data: https://tinyurl.com/y5evwkuw cross-treatment comparisons of DEG data: https://tinyurl.com/yyph68vc.

# Supplementary Information

# Acknowledgements

Support for these studies was provided by the National Institues of Health National Institute on Aging U01 AG046139, P30 AG066506, P50 AG047266. N Ertekin-Taner is also funded in part by National Institutes of Health National Institute on Aging RF1 AG051504, and R01 AG061796.

## Author Contributions

KN McFarland: data curation, formal analysis, investigation, visualization, methodology, and writing—original draft, review, and editing.
C Ceballos: investigation and methodology.
A Rosario: investigation and methodology.
TB Ladd: investigation and methodology.
BD Moore: investigation, methodology, and writing—review and editing.
G Golde: visualization.
X Wang: resources.
M Allen: resources.
N Ertekin-Taner: resources.
CC Funk: methodology.
M Robinson: methodology.
P Baloni: methodology.
N Rappaport: methodology.
P Chakrabarty: conceptualization, investigation, and methodology.
TE Golde: conceptualization, resources, supervision, funding acquisition, project administration, and writing—original draft, review, and editing.

## Conflict of Interest Statement

TE Golde is a cofounder of Lacerta Therapeutics and Andante Biologics Inc.

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
