## [Reviewer comments · Life Science Alliance]

Life Science Alliance

Microglia show differential transcriptomic response to A β peptide aggregates ex vivo and in vivo

Karen McFarland, Carolina Ceballos, Awilda Rosario, Thomas Ladd, Brenda Moore, Griffin Golde, Xue Wang, Mariet Allen, Nilufer Ertekin-Taner, Cory Funk, Max Robinson, Priyanka Baloni, Noa Rappaport, Paramita Chakrabarty, and Todd Golde

DOI: <https://doi.org/10.26508/lsa.202101108>

Corresponding author(s): Karen McFarland, University of Florida and Todd Golde, University of Florida

Review Timeline:	Submission Date:	2021-04-26
	Editorial Decision:	2021-05-20
	Revision Received:	2021-05-27
	Accepted:	2021-05-28

Scientific Editor: Shachi Bhatt

Transaction Report:

Please note that the manuscript was previously reviewed at another journal and the reports were taken into account in the decision-making process at Life Science Alliance.

May 20, 2021

RE: Life Science Alliance Manuscript #LSA-2021-01108-T

Karen N. McFarland
University of Florida
Neurology
1179 S Newell Dr, L3-100
Gainesville, FL 32610

Dear Dr. McFarland,

Thank you for submitting your revised manuscript entitled "Microglia show differential transcriptomic response to A β peptide aggregates ex vivo and in vivo". We would be happy to publish your paper in Life Science Alliance (LSA) pending minor text-based revisions mentioned below and final revisions necessary to meet our formatting guidelines.

For a brief overview, the manuscript was previously reviewed at a LSA partner journal, and the authors transferred the manuscript, along with the referee reports to LSA. At LSA, the manuscript and reviewer reports were assessed by both in-house editors and academic experts, who agreed that the dataset provided in the study was high quality and would be a valuable resource for the research community. The authors responses to the 2 points of concerns raised by the LSA editors (sent with the decision letter from the partner journal) were also sufficiently addressed by the authors. Thus, we would like to invite you to submit a final revision of this manuscript that includes the following minor edits:

- + We understand the argument that the authors have made in response to the concern from Rev 3 pt 2 about the 'cleanliness' of the oligomeric vs fibrillar preparations. We encourage the authors to discuss the caveats about this experiment in the manuscript, similar to what they included in the response
- + the minor revisions requested by the reviewers should be addressed in the revised manuscript

Along with the points listed above, we also encourage the authors to edit the following to meet the journal's formatting guidelines:

- please add a Summary Blurb/Alternate Abstract in our system
- please add Keywords and a Category for your manuscript in our system
- please add ORCID ID for secondary corresponding author-they should have received instructions on how to do so
- please consult our manuscript preparation guidelines <https://www.life-science-alliance.org/manuscript-prep> and make sure your manuscript sections are in the correct order
- please add your main, supplementary figure, and table legends to the main manuscript text after the references section
- please make sure the manuscript sections are aligned in accordance with LSA's formatting guidelines: please separate the Figure legends and Supplemental Figure legends into separate sections
- please add an Author Contributions section to your main manuscript text

- please add a conflict of interest statement to your main manuscript text
- we encourage you to revise the figure legends for figures 7 and S1 such that the figure panels are introduced in an alphabetical order
- there is a callout for Figure S3A and B although there is no legend for it nor the actual figure has been provided
- please add callouts for Figures 7B, C, and 8F-H to your main manuscript text
- please provide higher resolution higher quality images for the blots shown in Figure 1

A. FINAL FILES:

B. MANUSCRIPT ORGANIZATION AND FORMATTING:

****It is Life Science Alliance policy that if requested, original data images must be made available to**

the editors. Failure to provide original images upon request will result in unavoidable delays in publication. Please ensure that you have access to all original data images prior to final submission.**

The license to publish form must be signed before your manuscript can be sent to production. A link to the electronic license to publish form will be sent to the corresponding author only. Please take a moment to check your funder requirements.

Sincerely,

Shachi Bhatt, Ph.D.
Executive Editor
Life Science Alliance
<http://www.lsajournal.org>
Tweet @SciBhatt @LSAJournal

We have addressed the minor comments from the reviewers as requested:

Referee #2, Minor comment 1: Line 173, figure 3, authors cut the hierarchical tree at a height of 5.75. The authors should explain why they chose this value.

Response: Within the text, we have added our rationale for choosing to cut the tree at a height of 5.75.

Referee #2, Minor comment 2: Figure 7 and figure 8 are mis-referenced in the text. For instance, Figure 7 D in line 276 should rather be figure 7B.

Response: We thank the referees for their careful reading of our manuscript. This callouts in the manuscript for the panels in this figure have been corrected.

Referee #2, Minor comment 3: Line 372, "however we would note that both oA β and fA β induce massive changes in the transcriptome of microglia and certainly lend little credence to claims by some in the field that fA β is inert". Original publication should be cited.

Response: We have added the reference within the text as requested.

Referee #3, Minor comment 1: The introduction states that 'Yet, despite intensive study, the precise mechanism by which accumulation of A β aggregates trigger the degenerative phase of the disease is not well understood.' but this is not being illuminated in the current study.

Response: In the phrasing of this statement within the first paragraph of our introduction, we were merely setting the stage for the basis of these experiments.

Referee #3, Minor comment 2: While I understand that a colour coding is being used for modules, I find this confusing (or better: not informative) when only the colour coding is used in the main text: See lines 197-199: 'These modules include antiquewhite4, brown, coral1, darkseagreen4, honeydew1, lavenderblush3, lightcoral, lightcyan, lightcyan1, lightgreen, lightsteelblue1, orangered3, orangered4, saddlebrown, violet, white and yellow4.'

Response: WGCNA modules are originally identified by numbering the modules from largest to smallest based on the number of genes each contains. Typically, WGCNA modules are then named by color which makes for easier visualization. We are doubtful that referring to the modules by number instead of color will be any less confusing.

Referee #3, Minor comment 3: The figure legends are interspersed by Suppl Fig 1 and Table 2. Please move further down.

Response: We have separated the Supplemental figure legends and the table legends from the main figure legends.

Referee #3, Minor comment 4: I do not find Fig 1A very informative (nor aesthetically appealing). The filaments do not really look like beautiful Abeta filaments (with their typical periodicity). Fig. 1B needs to be complemented by negative contrast electron microscopy.

Response: We have removed figure 1A and placed the remaining portion of figure 1b as supplemental figure 1 (renaming the remaining figure within the text and legends). The preparations of oligomeric and fibrillar amyloid peptide that we use for these studies are ones that are commonly used in many other laboratories with previously published, detailed methodology for their preparation as referenced in the manuscript (Stine, 2003).

Referee #3, Minor comment 5: FPKM (only used in the panels of Fig 7) is a term/acronym that not everyone knows. It should be introduced in the legend/main text and then be explained.

Response: We have included the definition of FPKM in the figure legend for Figure 7 (now figure 6).

Please let us know if there is any additional information or changes that you require.

We look forward to the publication of our manuscript in Life Science Alliance.

Best,

Karen N McFarland, PhD

Todd E Golde, MD, PhD

May 28, 2021

RE: Life Science Alliance Manuscript #LSA-2021-01108-TR

Karen N. McFarland
University of Florida
Neurology
1179 S Newell Dr, L3-100
Gainesville, FL 32610

Dear Dr. McFarland,

Thank you for submitting your Research Article entitled "Microglia show differential transcriptomic response to A β peptide aggregates ex vivo and in vivo". It is a pleasure to let you know that your manuscript is now accepted for publication in Life Science Alliance. Congratulations on this interesting work.

DISTRIBUTION OF MATERIALS:

Again, congratulations on a very nice paper. I hope you found the review process to be constructive and are pleased with how the manuscript was handled editorially. We look forward to future exciting submissions from your lab.

Sincerely,

Shachi Bhatt, Ph.D.

Executive Editor

Life Science Alliance

<http://www.lsjournal.org>
